# Montreal Cognitive Assessment (MoCA) Norms for Older Patients with a Depressive Disorder

**DOI:** 10.3390/medsci13040312

**Published:** 2025-12-10

**Authors:** Myrthe E. Scheenen, Rob H. S. van den Brink, Styliani Konstantinidou, Astrid Lugtenburg, Jasmijn Spit, Gert-Jan Hendriks, Paul Naarding, Nathalie R. de Vent, Roy P. C. Kessels, Richard C. Oude Voshaar, Hans W. Jeuring

**Affiliations:** 1Department of Neurology & Alzheimer Center Groningen, University Medical Center Groningen, 9700 RB Groningen, The Netherlands; 2Department of Psychiatry, University Medical Center Groningen, University of Groningen, 9700 AB Groningen, The Netherlands; r.h.s.van.den.brink@umcg.nl (R.H.S.v.d.B.); r.c.oude.voshaar@umcg.nl (R.C.O.V.); h.w.jeuring@umcg.nl (H.W.J.); 3Department of Geriatric Medicine, University Medical Center Groningen, 9700 RB Groningen, The Netherlands; 4Graduate School of Medical Sciences, University of Groningen, 9700 AB Groningen, The Netherlands; s.konstantinidou.1@student.rug.nl; 5Department of Geriatric Psychiatry, GGZ Drenthe Mental Health Institute, 9401 LA Assen, The Netherlands; astrid.lugtenburg@ggzdrenthe.nl; 6Department of Geriatric Psychiatry Enschede, Mediant Mental Health Institute, 7546 RD Enschede, The Netherlands; ej.spit@mediant.nl; 7Pro Persona, Institute for Integrated Mental Health Care, 6500 HB Nijmegen, The Netherlands; g.hendriks@propersona.nl; 8Behavioural Science Institute, Radboud University, 6525 XZ Nijmegen, The Netherlands; 9Department of Psychiatry, Radboud University Medical Center, 6525 GA Nijmegen, The Netherlands; 10GGNet Apeldoorn, Department of Geriatric Psychiatry, 6525 CG Nijmegen, The Netherlands; p.naarding@ggnet.nl; 11Department of Psychology, University of Amsterdam, 1012 WX Amsterdam, The Netherlands; n.r.devent@uva.nl; 12Donders Institute for Brain Cognition and Behaviour, Radboud University, 6525 AJ Nijmegen, The Netherlands; 13Vincent van Gogh Institute for Psychiatry, 5802 EH Venray, The Netherlands; 14Radboudumc Alzheimer Center, Radboud University Medical Center, 6525 GA Nijmegen, The Netherlands

**Keywords:** depressive disorder, cognitive impairment, older adults, MoCA, precision

## Abstract

**Background**: Interpretation of cognitive performance in older patients with depression is challenging considering the association between late-life depression and (early-stage) neurodegenerative disease. The Montreal Cognitive Assessment (MoCA) is widely used to screen for mild cognitive impairment in community-dwelling older adults. **Objective**: The aim of the present study was to examine the need for and to develop dedicated MoCA norms for older people with depressive disorder. **Methods**: We used data from the Routine Outcome Monitoring for Geriatric Psychiatry & Science (ROM-GPS) study and the Advanced Neuropsychological Diagnostics Infrastructure (ANDI) database, which consisted of 859 patients with a depressive disorder according to DSM-5 criteria and 320 healthy controls, aged ≥60 years. Linear regression was used to examine the relationship between late-life depression and MoCA scores, adjusted for age, sex, and education. **Results**: The presence of a depressive disorder was associated with lower MoCA scores, and this effect was larger for persons with 12 years or less of education than for those with more education (B = −0.76 [95% CI −0.61; −0.91] vs. −0.53 [−0.36; −0.70]). Among depressed patients, depressive symptom severity was not associated with the MoCA score. Regression-based normative data for the MoCA were computed and adjusted for age, education, sex, and type of depressive disorder. **Conclusions**: Our findings demonstrate that depressive disorder, but not symptom severity within depression, is associated with lower MoCA scores. Clinical interpretation of MoCA scores in depressed older persons can be facilitated by using MoCA reference tables stratified by age, sex and level of education.

## 1. Introduction

Depression is the leading cause of emotional suffering in older adults [1], with 10–15% of those over 65 experiencing clinically relevant depressive symptoms [2], and nearly 5% meeting criteria for major depressive disorder (MDD). Unipolar depressive disorders (UDD), including MDD and persistent depressive disorder (PDD), involve significant mood disruptions that impair social and occupational functioning [3]. These disorders can also affect cognitive abilities [4,5], and conversely, cognitive decline can negatively impact mood, particularly in the elderly [6]. Differentiating between cognitive symptoms caused by depression and those due to early neurodegenerative diseases poses a clinical challenge in geriatric psychiatry [7].

While debate exists about the value of cognitive screening [8,9], it remains standard clinical practice. These screenings help to identify mild cognitive impairment (MCI), a transitional stage between normal aging and dementia [10], and determine whether a patient requires further neuropsychological testing or can begin treatment for UDD [7]. Traditionally, the Mini-Mental State Examination (MMSE) has been widely used [11], but the Montreal Cognitive Assessment (MoCA) has gained traction due to better sensitivity for detecting MCI [12,13]. A MoCA cut-off score below 26 was initially proposed to differentiate between healthy aging, MCI, and Alzheimer’s dementia, showing high sensitivity and specificity in early studies [12]. However, subsequent research has reported reduced accuracy, with increased risks of false positives and negatives [14,15]. To address this shortcoming, Kessels and colleagues (2022) developed regression-based normative data adjusted for age, education, and sex, emphasizing the need for individualized interpretation rather than use of a universal cut-off [16]. Despite its frequent use in geriatric psychiatry, little is known about how the MoCA performs in older adults with UDD [7].

Research suggests that specific clinical settings and patient characteristics may further influence MoCA performance [7,14,17], potentially requiring additional adjustments to improve accuracy. Although UDD is linked to cognitive deficits, particularly in processing speed, memory, and executive functioning, the effects are typically modest, making significant drops in MoCA scores less likely [18,19]. However, studies have shown that depressive symptoms can lead to lower scores on both the MMSE and MoCA [20,21], and a study among patients with late-life depression demonstrated false positive cases of MCI by using the MoCa [7]. Most studies interpret lower scores on screening tools as being due to motivational deficits associated with depression rather than true cognitive decline, but the relationship between depression and cognitive decline remains complex.

Several studies show that depression is associated with a higher risk of developing dementia, but clear causation has not been established. Some studies suggest that depression might be caused by perceived deterioration in cognition [22], others that it might be a prodromal symptom of dementia [23]. There are, however, also clear indications that depression influences cognition regardless of cognitive decline. For example, in several studies among younger adults living with HIV/AIDS, it was found that patients with MDD scored significantly poorer on the MoCA than patients without MDD, suggesting that the depressive disorder itself negatively influences the MoCA score [24,25,26]. Moreover, a study on depressive symptoms on MoCA scores in a memory clinic setting found a negative impact of depressive symptoms on MoCA scores; follow-up showed that the majority of returning patients (95%) had no progression to a neurodegenerative disease [21]. Furthermore, a large cross-sectional study by Wiels and colleagues recently showed that the presence and severity of depression were not consistently associated with amyloid pathology [27]. On the other hand, more severe depression has consistently been linked to poorer MoCA performance in patients with MDD [6,20].

In summary, it has already been found that a more nuanced interpretation that accounts for demographic and clinical variables is likely necessary when using the MoCA [15,16,28]. However, it has not yet been established if MoCA scores are negatively influenced by depression in such a way that dedicated norms are needed to further improve reliability for use in geriatric psychiatry. Therefore, the primary aim of this study is to assess whether UDD and severity of depression confound the MoCA scores in older adults. We hypothesize that both the presence of a UDD and higher depression severity negatively influence MoCA results. If confirmed, we aim to develop adjusted reference tables for MoCA scores tailored to older adults with UDD, incorporating age, sex, education, and depression characteristics, using a regression-based approach. This may ultimately contribute to more personalized diagnostics, reduce misdiagnosis, and optimize healthcare resource use.

## 2. Methods

### 2.1. Study Design

Data from two study samples were combined to study determinants of the MoCA score: a clinical and a non-clinical sample.

The clinical sample consisted of 859 patients of 60 years or older with a depressive disorder who were referred to one of the eight outpatient clinics for specialized geriatric mental healthcare that participated in the Routine Outcome Monitoring for Geriatric Psychiatry & Science (ROM-GPS) study [29]. All clinics had a partly standardized intake procedure, including the assessments used in the present study, administered by trained research nurses. Excluded from our clinical sample were patients with an established neurodegenerative disorder, a (history of a) bipolar or psychotic disorder, or insufficient mastery of the Dutch language. The ROM-GPS study was approved by the Medical Ethics Committee of the University Medical Center Groningen on the 14th of June 2014 (NL47717.042.14) and was registered on the International Clinical Trial Registry Platform (ICTRP Number NTR6874).

The non-clinical sample consisted of 320 adults of 60 years or older whose MoCA score was included in the Advanced Neuropsychological Diagnostics Infrastructure (ANDI) database (www.andi.nl) [30]. All studies were approved by local ethics committees. Exclusion criteria were any condition with profound impact on the brain or cognitive health beyond normal aging, such as psychiatric disorders, neurological conditions, and substance use disorders. The MoCA data of the database were previously used to construct regression-based normative data for the MoCA for adults aged 18–91 years [16].

### 2.2. Outcome Measure

Outcome measure was the Montreal Cognitive Assessment (MoCA), which is a brief cognitive screening tool developed with the aim of detecting MCI [12]. Participants in both the clinical and non-clinical samples were administered the authorized paper-and-pencil Dutch version of the Montreal Cognitive Assessment version 7.1.

The MoCA sum score ranges from 0 to 30, with higher scores indicating better cognitive functioning. For clinical use, this sum score needs to be corrected if patients had 12 years or less of education, in which case one point is added, provided the score does not exceed 30. Here, we use the MoCA sum score before correction for education. For all participants of the clinical sample and those of the non-clinical sample for whom item scores were available, subscores were also calculated for the following parts of the MoCA indicated in the scoring manual: Visuospatial/Executive, Naming, Attention, Language, Abstraction, Delayed Recall, and Orientation.

### 2.3. Main Determinants

#### 2.3.1. Depressive Disorder

Presence of a current MDD or PDD according to DSM-5 criteria was established in the clinical sample with the Dutch version of the Mini International Neuropsychiatric Interview Plus (MINI-Plus) [31], a semi-structured psychiatric interview.

#### 2.3.2. Depression Severity

Severity of depressive symptoms was assessed in the clinical sample with the Inventory of Depressive Symptomatology (IDS-SR) 30-item self-report questionnaire [32]. The sum score ranges from 0 to 60 and classifies depression severity as none (score 0–12), mild (13–24), moderate (25–37), severe (38–47), or very severe (≥48).

### 2.4. Control Variables

#### 2.4.1. Level of Education

For the clinical sample, level of education was assessed as ≤12 years of education versus >12 years, as indicated in the MoCA manual. For the non-clinical sample, it was assessed in seven levels based on the Dutch educational system [33], which were divided for this study into the lower five levels (equivalent to ≤12 years of education) [16] versus the two higher levels (which require more than 12 years of education).

#### 2.4.2. Comorbid Mental Disorders

In the clinical sample, the MINI-Plus was used to assess the following comorbid mental disorders according to DSM-5 criteria [3]: anxiety disorders, somatic symptoms or related disorders, post-traumatic stress disorder, and obsessive–compulsive disorder.

### 2.5. Analyses

Determinants of the MoCA score were examined by linear regression. The standardized data-handling procedure for the ANDI database [30] was followed to check the assumptions of homoscedasticity and normal distribution of residuals and to transform the MoCA outcome score if necessary.

The association between depressive disorder and MoCA scores was studied in the combined sample, controlled for the control variables (sex, age, and education) and any significant (first, second or third order) interaction between a depressive disorder and control variables. We further explored whether effects found were seen on all or only specific—untransformed—subscores of the MoCA.

The association between characteristics of a depressive disorder and the MoCA sum score was studied in the clinical sample. Effects of type of depressive disorder (MDD or PDD), depression severity, presence of a comorbid mental disorder, the control variables, and significant first-order interactions between these determinants were studied.

In all analyses, the effect size of associations with the MoCA outcome was assessed by partial eta squared (η_p_^2^), which reflects the proportion of variance in the (transformed) MoCA outcome explained by the determinant. The effect of the determinant on the untransformed MoCA score was investigated by the mean difference in back-transformed predictions of the MoCA score between different levels of the determinant.

### 2.6. Analyses for Reference Tables

If a significant influence of a depressive disorder or depression characteristics was found, we composed MoCA score reference tables for use in older adults with a unipolar depressive disorder (MDD or PDD), stratified by age, sex, educational level, type of depressive disorder, depression severity and presence of a comorbid mental disorder, if indicated. Using the regression-based normative approach, described by Kessels and colleagues [16], we computed percentile distributions of the residual scores corresponding to each MoCA score minus the regression-based expected score for the specific stratum. These expected scores were based on the multivariate model for the clinical sample but included significant determinants only. For strata defined by continuous determinants, among others, the expected score for the stratum was based on the midrange values of these determinants. For each stratum, we first calculated residual scores for all possible MoCA scores minus the stratum-specific expected score. Subsequently, we converted these residual scores—and hence corresponding MoCA scores—into percentile scores, using the percentile distribution of residual scores from subject-specific expected scores in the total clinical sample. We tested whether it is justified to assume that the variance of residuals is constant across the values of the determinants using the Modified Breusch–Pagan test for heteroscedasticity [34].

## 3. Results

### 3.1. Sample Characteristics

Out of 1047 patients initially eligible for the clinical sample, 188 (18.0%) were excluded due to neurodegenerative disorders (2.3%), bipolar disorder history (6.9%), psychotic disorder history (3.0%), insufficient Dutch language proficiency (0.7%), or missing MoCA data (5.2%). This left 859 patients (82.0%) in the clinical sample. For the non-clinical group, data from 320 participants were drawn from the ANDI database, with MoCA subscores available for 53 individuals. These 53 participants did not differ from the full non-clinical sample in terms of age, sex, education, or MoCA total score.

As shown in Table 1, the clinical and non-clinical groups were comparable on age, sex, and education. However, the clinical group scored lower on the MoCA, both before and after correcting for education, and showed greater score variance (Levene’s test: F = 18.52, *p* < 0.001 and F = 19.66, *p* < 0.001). Thirty extreme outliers were removed (23 clinical, 7 non-clinical), and the MoCA scores were normalized (raised to the power of 2.85), as advised on the ANDI website for the MoCA (www.andi.nl), and subsequently standardized.

### 3.2. Association with Depressive Disorder

Univariate analysis (Table 2, top) revealed negative associations between MoCA scores and depressive disorder status, older age, and lower education, whereas no association was found with sex. Among tested interactions, only the interaction between depressive disorder and education level was significant (F = 6.21, *p* = *0*.013, ηp^2^ = 0.005). This interaction was included in the multivariate model (Table 2, bottom).

Multivariate analysis confirmed significant associations between MoCA scores and all control variables, including sex, and the depressive disorder × education interaction. Patients with depressive disorder and ≤12 years of education had a greater MoCA score reduction (−2.68 points, ηp^2^ = 0.082) than those with more education (−1.44 points, ηp^2^ = 0.032). Likewise, higher education had a stronger positive association with MoCA scores in the depressed group (+2.72 points, ηp^2^ = 0.125) than in the non-depressed group (+1.48 points, ηp^2^ = 0.026). Age had a negative association with MoCA scores (−1.58 points between youngest and oldest halves), and women scored slightly higher than men (+0.10 points).

Exploratory subscore analyses (Appendix A, Table A1) showed that depressive disorder was negatively related to all MoCA subscores except Naming, with the strongest relation found on the visuospatial/executive subscore (ηp^2^ = 0.20).

### 3.3. Clinical Characteristics of Depressive Disorder

Table 3 shows the association of depressive disorder subtype, severity, and comorbid mental disorders and MoCA scores within the clinical group. Univariate analyses revealed no significant effects. However, multivariate analysis indicated slightly higher MoCA scores for patients with MDD compared to PDD (+0.28 points) and a small positive effect for women over men (+0.24 points). Education’s effect again varied by age: patients with >12 years of education had a greater benefit if older (+2.95 points) compared to younger patients (+2.06 points). Conversely, age’s negative impact was stronger in those with ≤12 years of education (−0.05 points per year, ηp^2^ = 0.076) versus those with more education (−0.03 points per year, ηp^2^ = 0.021).

### 3.4. Percentile Scores for the MoCA

The multivariate regression model, including significant determinants only, on which the expected MoCA scores were based, is provided in the Appendix (Table A2). The Modified Breusch–Pagan test for heteroskedasticity for this regression model showed no indication of differences in variance of the residual scores over the values of the determinants (Chi-square = 0.48, df = 1, *p* = 0.487). The reference tables, providing percentile scores for observed MoCA scores (before correction for educational level) for the different combinations of the determinants, are presented below in Table 4.

Table 5 shows examples of MoCA percentile scores that were adjusted for sex, age, educational level, and type of depressive disorder (DD) based on Table 4. These examples may help clinicians contextualize individual MoCA results within relevant reference groups.

For example, (i) a 76-year-old woman with MDD, 12 years of education, and a MoCA score of 16 scored at the 7th percentile—meaning that 7 percent of the reference group scored the same or lower than her (more likely attributable to neurodegenerative disease); (ii) an 85-year-old man with MDD, 11 years of education, and a MoCA score of 22 scored at the 68th percentile—meaning that 68% scored the same or lower than him (more likely attributable to depressive disorder). 

## 4. Discussion

The primary finding of this study is that the presence of a unipolar depressive disorder (UDD), as defined by DSM criteria, predicts lower Montreal Cognitive Assessment (MoCA) scores in older adults, regardless of depression severity or psychiatric comorbidity. This emphasizes the need for personalized reference tables to accurately interpret MoCA scores in geriatric depression. Therefore, we present regression-based normative data for MoCA scores, adjusted for age, education, sex, and type of UDD (major depressive disorder [MDD] or persistent depressive disorder [PDD]). These reference values are recommended for clinical use in geriatric psychiatry.

Our findings align with prior research showing that depressive disorders are associated with cognitive impairments [35], that MDD is linked to worse MoCA performance in various clinical groups, and that lower MoCA scores are reported in small geriatric psychiatry samples with depression [7,20]. This may be due to motivational deficits caused by depression, particularly apathy—a common feature in late-life depression that affects episodic memory and processing speed [7,20]. Sözeri-Varma and colleagues (2019) found that older patients with depression had higher apathy levels and lower MoCA scores than controls, with both apathy and depression severity predicting cognitive deficits [36]. Psychomotor slowing, another hallmark of depression, may also play a role in reduced MoCA scores [37]. While slowing is part of normal cognitive aging, it increases the risk for depressive and anxiety symptoms and contributes to impaired executive functioning [38].

Consistent with recent findings by Kessels and colleagues (2022) [16], we observed associations between MoCA scores and demographic variables: age, sex, and education. Younger participants and those with more education scored higher on the MoCA, which has been supported by prior research [39,40,41,42,43]. Although sex-related cognitive differences are generally small [16,44], women in our study outperformed men on the MoCA in both depressed and non-depressed groups. Lower education levels were significantly associated with reduced MoCA performance, especially among the older half of the depressed sample.

The protective effect of education on cognitive performance is well-established and attributed to greater cognitive reserve, a concept reflecting the brain’s resilience and adaptability [45,46]. Cognitive reserve is influenced not only by education but also by intelligence, occupational complexity, and lifestyle [35]. Older adults with higher education are more likely to have engaged in cognitively and socially stimulating activities, which may delay cognitive decline [47]. In contrast, depression can reduce activity levels [1], especially among those with lower education, which may amplify cognitive vulnerabilities.

Notably, the association between UDD and MoCA scores was independent of psychiatric comorbidities and, contrary to previous findings [36], unrelated to depression severity. We did not replicate Sözeri-Varma et al.’s observation that more severe depression lowers MoCA performance. However, patients with PDD showed somewhat greater MoCA impairment than those with MDD, possibly due to the chronicity of PDD symptoms [48]. All MoCA subdomains were associated with lower scores in depressed patients, except Naming. This challenges the clinical utility of focusing on MoCA subscores for differential interpretation in depressed older adults.

### 4.1. Limitations

This study has several limitations. Firstly, because our study did not have specific data on medication use at the time of the MoCA measurement, we cannot rule out that certain psychotropic medications influenced cognitive performance. However, although our study lacks power to distinguish between different (combinations of) medications, a sensitivity analysis investigating the influence of the use of psychotropic medication the year before intake showed no changes in the effect of a depressive disorder on MoCA scores. Secondly, although individuals with diagnosed neurodegenerative disorders were excluded, we did not conduct full neurocognitive assessments (comprehensive neuropsychological testing and/or MRI). Some participants may have had subclinical cognitive impairments indicative of early neurodegenerative disease. Although we did try to minimize this risk by excluding cases with abnormally low MoCA scores, this could have introduced a confounding bias. Additionally, we did not exclude those with other neurological conditions, such as stroke or transient ischemic attacks, which may have confounded cognitive performance. However, with regard to all above-mentioned points, our approach reflects a real-world clinical population, where complex medical histories are common. Limiting the sample to “pure” cases would compromise external validity. Lastly, we used the Dutch version of the MoCA. Caution is warranted when applying normative data across different language versions, as translation can affect performance patterns [49]. Nonetheless, our findings offer valuable clinical insights.

### 4.2. Clinical Implications

In summary, UDD in older adults is associated with significantly lower MoCA scores, especially among those with lower educational attainment. This relationship is more pronounced in the oldest patients with fewer years of education. Using MoCA to screen for cognitive impairment in depressed older adults raises the risk of false positive cases (i.e., mistaking depression-related deficits for early dementia) and false negative cases (attributing cognitive symptoms solely to depressive disorder). Both outcomes can result in emotional distress and financial burden.

Therefore, it is crucial to use individualized interpretation strategies when applying the MoCA in clinical settings. The availability of age-, sex-, and education-adjusted normative data represents a significant improvement over a fixed cut-off score approach [15,16,28]. Building on this work, we have now created personalized MoCA reference tables for older adults with UDD, adjusted for demographic and clinical characteristics. These tools can enhance diagnostic accuracy and support clinical decision-making in geriatric psychiatry. By improving the interpretability of MoCA scores in older adults with depressive disorder, we hope to aid clinicians in providing better, more targeted care and avoiding the pitfalls of misdiagnosis.

## Figures and Tables

**Table 1 medsci-13-00312-t001:** Characteristics of the study samples.

Characteristic	Clinical Sample (*n* = 859)	Non-Clinical Sample (*n* = 320)	Difference		
			Χ^2^/t	df	*p*
Sex, female, *n* (%)	529 (61.6%)	178 (55.6%)	3.45	1	0.063
Age, years, mean (SD)	70.3 (7.1)	69.7 (6.8)	1.41	1177	0.159
Age group					
- 60–64 year	202 (23.5%)	87 (27.2%)			
- 65–69 year	226 (26.3%)	85 (26.6%)			
- 70–74 year	198 (23.1%)	69 (21.6%)			
- 75–79 year	133 (15.5%)	47 (14.7%)	2.17	4	0.704
- ≥80 year	100 (11.6%)	32 (10.0%)			
Education, more than 12 years, *n* (%)	377 (43.9%)	136 (42.5%)	0.18	1	0.669
MoCA score					
- before correction for education *, mean (SD)	23.1 (3.9)	25.4 (3.2)	10.48	697	<0.001
- after correction for education *, mean (SD)	23.7 (3.7)	26.0 (3.0)	10.91	700	<0.001
Major depressive disorder, *n* (%)	473 (55.1%)				
Persistent depressive disorder, *n* (%)	386 (44.9%)				
Severity of depression, IDS, mean (SD)	38.1 (11.2)				
Severity of depression level, IDS, *n* (%)					
none/mild	105 (12.2%)				
moderate	312 (36.3%)				
severe	264 (30.7%)				
very severe	144 (16.8%)				
missing	34 (4.0%)				
Comorbid mental disorders					
anxiety disorder ^#^, *n* (%)	356 (41.4%)				
somatic symptom or related disorder ^$^, *n* (%)	185 (21.5%)				
post-traumatic stress disorder, *n* (%)	83 (9.7%)				
obsessive–compulsive disorder, *n* (%)	62 (7.2%)				
any, *n* (%)	471 (54.8%)				

* T-test with equal variances not assumed. # Generalized anxiety disorder, panic disorder, agoraphobia, or social anxiety disorder. $ Somatic symptom disorder, or illness anxiety disorder.

**Table 2 medsci-13-00312-t002:** Determinants of transformed MoCA score in older adults (N = 1149).

Determinant	Regression Model for Transformed MoCa Scores ^#^						Back-Transformed Predictions
	B	95% CI	t	df	*p*	η_p_^2^	Mean Difference
** *Univariate* **							
Depressive disorder	−0.66	−0.54; −0.79	10.44	1147	<0.001	0.087	−2.07
Sex, female	0.03	−0.09; 0.15	0.50	1147	0.923	0.000	0.10
Age, years	−0.04	−0.03; −0.05	10.19	1147	<0.001	0.083	−1.55 ^&^
Education, more than 12 years	0.72	0.61; 0.83	13.00	1147	<0.001	0.128	2.31
** *Multivariate* **							
Depressive disorder	−0.76	−0.61; −0.91	10.13	1143	<0.001	0.082	−2.68
Sex, female	0.22	0.12; 0.32	4.22	1143	<0.001	0.015	0.10
Age, years	−0.03	−0.03; −0.04	9.48	1143	<0.001	0.073	−1.58 ^&^
Education, more than 12 years	0.54	0.35; 0.73	5.55	1143	<0.001	0.026	1.48
Depressive disorder * Education ^$^	0.23	0.01; 0.45	2.03	1143	0.043	0.004	1.10

# MoCA score before correction for education, transformed to power advised on ANDI website and standardized. & Oldest half of respondents (70 years or older) compared with youngest half (younger than 70). * Indicate the interaction term between two. $ Effect of depressive disorder in group with education of more than 12 years: −0.53 (95% CI: −0.36; −0.70), [back-transformed: −1.44], t = 6.19, df = 1143, *p* = <.001, η_p_^2^ = 0.032. Effect of education of more than 12 years in group with depressive disorder: 0.77 (95% CI: 0.65; 0.89), [back-transformed: 2.72], t = 12.77, df = 1143, *p* < 0.001, η_p_^2^ = 0.125.

**Table 3 medsci-13-00312-t003:** Determinants of transformed MoCA score in older adults with a depressive disorder (N = 836).

Determinant	Regression Model for Transformed MoCa Scores ^#^						Back-Transformed Predictions
	B	95% CI	t	df	*p*	η_p_^2^	Mean Difference
** *Univariate* **							
Depressive disorder, Major DD ^$^	0.08	−0.06; 0.21	1.08	834	0.280	0.001	0.25
Severity of depression, IDS-score	0.00	−0.01; 0.00	0.77	802	0.439	0.001	−0.14 *
Comorbid mental disorder, any	0.00	−0.13; 0.14	0.04	834	0.965	0.000	0.01
Sex, female	0.07	−0.07; 0.21	0.94	834	0.350	0.001	0.22
Education, more than 12 years	0.83	0.71; 0.96	13.08	834	<0.001	0.170	2.70
Age, years	−0.04	−0.05; −0.04	9.30	834	<0.001	0.094	−1.66 ^&^
** *Multivariate* **							
Depressive disorder, Major DD ^$^	0.20	0.08; 0.33	3.17	796	0.002	0.012	0.28
Severity of depression, IDS-score	0.00	−0.01; 0.00	1.19	796	0.236	0.002	−0.29 *
Comorbid mental disorder, any	−0.06	−0.19; 0.07	0.89	796	0.372	0.001	−0.01
Sex, female	0.25	0.12; 0.37	3.90	796	<0.001	0.019	0.24
Education, more than 12 years ^^^	0.58	0.37; 0.80	5.32	796	<0.001	0.034	2.06
Age, years ^@^	−0.05	−0.04; −0.06	8.08	796	<0.001	0.076	−1.75 ^!^
Age * Education (>12 years)	0.02	0.00; 0.04	2.12	796	0.034	0.006	-

# MoCA score before correction for education, transformed to power advised on ANDI website and standardized. $ Compared to patients with a persistent depressive disorder. * Half of respondents with severest depressions compared with half with mildest depressions (median split in IDS-score ≥ 39 vs. <39). & Oldest half of respondents (70 years or older) compared with youngest half (younger than 70). ^ Effect of Education of more than 12 years in people who are 60 years old, and back-transformed difference in youngest half of respondents (younger than 70 years). Back-transformed difference in oldest half (70 years or older) is 2.95. @ Effect of age (per year) in people with 12 years or less of education. Effect of age in people with more than 12 years of education is −0.03 (95% CI: −0.02; −0.04), t = 4.16, df = 796, *p* < 0.001, η_p_^2^ = 0.021. ! Back-transformed difference between oldest half of respondents (70 years or older) compared with youngest half (younger than 70 years) in people with 12 years or less of education. Back-transformed difference in people with more than 12 years of education is −0.87.

**Table 4 medsci-13-00312-t004:** Reference table (percentile scores for the MoCA total score before correction for education, stratified by type of depressive disorder, age group, educational level, and sex).

Disorder	*Major Depressive Disorder*																			
Age *	60–64 Years				65–69 Years				70–74 Years				75–79 Years				≥80 Years			
Education	≤12 Year		>12 Year		≤12 Year		>12 Year		≤12 Year		>12 Year		≤12 Year		>12 Year		≤12 Year		>12 Year	
Sex	M	F	M	F	M	F	M	F	M	F	M	F	M	F	M	F	M	F	M	F
MoCa Score ^#^																				
30	100	100	100	100	100	100	100	100	100	100	100	100	100	100	100	100	100	100	100	100
29	99	97	92	87	100	99	94	90	100	100	96	93	100	100	97	95	100	100	98	96
28	96	92	82	75	98	95	86	80	99	98	89	84	100	99	92	87	100	100	94	90
27	89	83	69	59	93	89	74	65	96	93	79	71	98	96	83	76	100	98	87	81
26	80	72	54	46	86	79	59	51	91	85	65	56	94	91	71	61	97	94	76	66
25	66	57	42	32	76	66	47	37	83	76	53	44	89	82	57	49	93	88	63	54
24	54	46	28	19	63	54	35	24	73	63	40	30	80	72	46	36	86	80	51	42
23	44	35	16	11	53	44	21	14	60	52	28	18	70	60	34	23	78	69	39	29
22	33	22	9	4	42	32	14	7	51	42	17	11	58	50	21	14	68	58	28	19
21	21	14	4	1	32	21	6	2	41	31	11	5	50	41	14	8	58	50	18	12
20	14	9	1	0	21	14	2	0	32	21	5	1	41	31	8	3	50	41	12	6
19	9	4	0	0	15	9	1	0	22	15	1	0	33	22	4	1	42	32	6	2
18	5	1	0	0	10	5	0	0	15	9	0	0	24	15	1	0	35	23	2	1
17	2	0	0	0	6	1	0	0	12	6	0	0	18	12	0	0	27	17	1	0
16	1	0	0	0	3	1	0	0	7	2	0	0	14	7	0	0	20	14	0	0
15	0	0	0	0	1	0	0	0	5	1	0	0	10	4	0	0	15	9	0	0
14	0	0	0	0	1	0	0	0	2	0	0	0	7	2	0	0	13	6	0	0
≤13	0	0	0	0	0	0	0	0	1	0	0	0	5	1	0	0	10	4	0	0
	** *Persistent depressive disorder* **																			
30	100	100	100	100	100	100	100	100	100	100	100	100	100	100	100	100	100	100	100	100
29	100	99	95	92	100	100	96	94	100	100	98	96	100	100	98	97	100	100	100	98
28	98	95	88	82	99	98	91	85	100	99	93	89	100	100	94	92	100	100	96	94
27	93	89	77	67	96	93	81	74	98	96	85	78	99	98	88	82	100	99	91	86
26	85	79	62	54	91	85	68	58	94	90	74	64	97	94	78	70	98	97	82	75
25	75	66	50	40	82	74	54	46	88	82	59	51	93	88	65	56	96	93	72	61
24	62	54	37	27	72	62	43	33	80	72	48	39	86	79	54	45	91	85	58	50
23	51	43	24	15	59	51	31	20	69	59	37	27	77	69	43	33	85	77	48	39
22	42	32	15	9	50	41	19	13	58	50	25	16	67	57	31	21	76	66	37	27
21	31	20	9	4	40	31	13	6	50	40	15	9	57	49	20	14	66	57	27	17
20	21	14	4	1	31	20	6	2	40	31	10	4	49	40	14	7	57	49	17	11
19	15	9	1	0	21	14	2	0	32	21	5	1	41	31	8	3	50	41	12	6
18	9	4	0	0	15	9	1	0	23	15	2	0	33	23	4	1	43	33	7	2
17	6	1	0	0	11	6	0	0	17	11	1	0	26	17	1	0	36	25	4	1
16	2	1	0	0	7	2	0	0	14	7	0	0	20	13	0	0	29	19	1	0
15	1	0	0	0	4	1	0	0	9	4	0	0	15	9	0	0	23	15	1	0
14	0	0	0	0	2	0	0	0	6	2	0	0	13	6	0	0	19	13	0	0
≤13	0	0	0	0	1	0	0	0	4	1	0	0	9	4	0	0	15	9	0	0

* For the age groups, the stratum-specific expected scores were based on the midrange values 62.5, 67.5, 72.5 and 77.5 for the groups 60–64 years, 64–69 years, 70–74 years and 75–79 years, respectively, and 82.5 for group 80 years or older. # MoCA total score before correction for education.

**Table 5 medsci-13-00312-t005:** Examples of percentile scores for the MoCA total score, adjusted for sex, age, years of educational, and type of depressive disorder (DD).

Sex	Age	Education	Type of DD	MoCA Score	Percentile Score (Reference Group)
Female	76	12	MDD	16	7th
Male	80	10	PDD	15	23rd
Female	62	14	PDD	25	40th
Male	85	11	MDD	22	68th

MDD: Major Depressive Disorder; PDD: Persistent Depressive Disorder.

## Data Availability

Researchers interested in checking or using the data of the ROM-GPS project, can contact the principal investigator R.C. Oude Voshaar (r.c.oude.voshaar@umcg. nl). Data for future studies can be obtained by submitting a proposal for collaborative research with the ROM-GPS researchers.

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
