# Peer review of "Montreal Cognitive Assessment (MoCA) Norms for Older Patients with a Depressive Disorder"

_medsci, 2025, doi:10.3390/medsci13040312_

Round 1
Reviewer 1 Report
Comments and Suggestions for Authors
This manuscript reports on the relationship between depression in older adults and its severity and scores obtained with the MoCA, a major tool for detecting mild and major neurocognitive disorders.
The subject is relevant because late-onset depression is often associated with neurocognitive disorders, and depression is known to influence cognitive function and its assessment.
To this end, the authors analyzed in detail the MoCA results obtained in a population consisting of a cohort of 859 elderly people with well-documented depression and 320 adults over the age of 60 selected from a Dutch database. The authors found a significant association between depression severity score and MoCA score. This link is significant, regardless of age, gender, and education level in the multivariate analysis.
Key points
As the authors pointed out in the “Limitations” section of the discussion, even though people with neurodegenerative diseases were excluded, it is possible that people with mild neurocognitive disorders (NCD) or mild cognitive impairment (MCI) may have been included in the cohort of people with depression. Given that MCI or mild TNC are known to decrease MoCA scores, they may have introduced a source of confounding bias. This important point should be more clearly stated and discussed in the limitations section. In addition, the authors could have performed a sensitivity analysis by excluding individuals who reported subjective memory impairment on the IDS-SR scale that comprise an item related to memory complaints. The absence of self-reported memory symptoms reasonably rules out individuals with MCI. If the relationship remains significant, this would make the results more convincing.
The authors did not take into account the potential role of psychotropic drugs on MoCA scores in the cohort of patients suffering from depression. Could they provide the proportions of patients who were taking antidepressants, benzodiazepines, or other psychotropic drugs at the time of the MoCA test? Certain drugs such as benzodiazepines, tricyclic antidepressants, or possibly mirtazapine could influence certain features of cognitive function. This could also introduce a certain bias. Once again, a sensitivity analysis excluding patients exposed to drugs that could influence cognitive testing could prove very useful.
Minor point
The correlations between depression and the MoCA sub-items are very interesting and could be included in the main body of the manuscript rather than in the supplementary material.
Reviewer 2 Report
Comments and Suggestions for Authors
The main strength of this article is the samples that were accessed.
I have three overall points to make-
(1) The authors are using a predictive design and predictive analyses, but they use the language of causation (e.g., affected, line 45; influence, line 162; effect, line 198; impact, line 301). Words such as "predict" or "are associated with differences" are more appropriate.
(2) The authors did not really disentangle depression from cognitive decay. Each may cause the other. Both might influence motivation to perform. Are participants' scores lower because they are depressed or are they depressed because they note a decrease in functioning? A little more discussion of this issue in the introduction (a paragraph) would be welcome.
(3) Ethics: was the use that the authors made of the data covered under some original ethics approval and participant consent forms? This should be mentioned in the text (I note that it does appear at the end of the article.)
Here are some minor points, mostly questions:
a. We know the two groups form which data originated (ROM-GPS and ANDI) but might the way in which data were collected have led to any biases? Were members from one group only in search of a diagnosis for their issues?
b. What was the time span for data collection, and were the two sources collected at roughly the same times?
c. Data normalization is mentioned well before it is explained (line 175).
d. I found the percentage of cases with comorbidities (>50%) daunting. Is this typical?
e. Line 251: I am sure the cases mentioned in Table 4 were not actually random (unless they were, in which case a random number generator would have to have been employed).
f. Since the primary reason for the authors writing the article is the provision of new norms, I would like to see Appendix A in the main part of the article.
Round 2
Reviewer 1 Report
Comments and Suggestions for Authors
The authors responded satisfactorily to the points raised and the manuscript was amended accordingly.
However, surprisingly, they added several terms relating to prediction (p6 and p7) without this being requested. This study cannot make any predictions. The authors should use more statistically appropriate terms, such as association.
Author Response
Comments on revisions Reviewer 1
Comment: The authors responded satisfactorily to the points raised and the manuscript was amended accordingly.
However, surprisingly, they added several terms relating to prediction (p6 and p7) without this being requested. This study cannot make any predictions. The authors should use more statistically appropriate terms, such as association.
Response: The reviewer is right to point out that we changed some terms related to prediction. This was requested by another reviewer who found the original wording to be implying causation. We agree with the reviewer that the use of terms relating to prediction is not appropriate, so we have made the suggested changes accordingly.